# Chemical Constituents of the Deep-Sea-Derived *Penicillium citreonigrum* MCCC 3A00169 and Their Antiproliferative Effects

**DOI:** 10.3390/md20120736

**Published:** 2022-11-24

**Authors:** Zheng-Biao Zou, Gang Zhang, Yu-Qi Zhou, Chun-Lan Xie, Ming-Min Xie, Lin Xu, You-Jia Hao, Lian-Zhong Luo, Xiao-Kun Zhang, Xian-Wen Yang, Jun-Song Wang

**Affiliations:** 1Center for Molecular Metabolism, School of Environmental and Biological Engineering, Nanjing University of Science and Technology, 200 Xiaolingwei Street, Nanjing 210094, China; 2Key Laboratory of Marine Genetic Resources, Third Institute of Oceanography, Ministry of Natural Resources, 184 Daxue Road, Xiamen 361005, China; 3Xiamen Key Laboratory of Marine Medicinal Natural Products Resources, Xiamen Medica College, 1999 Guankouzhong Road, Xiamen 361023, China; 4School of Pharmaceutical Sciences, Xiamen University, South Xiangan Road, Xiamen 361102, China

**Keywords:** deep-sea, fungus, *Penicillium solitum*, anti-tumor, apoptosis

## Abstract

Six new citreoviridins (citreoviridins J–O, **1**–**6**) and twenty-two known compounds (**7**–**28**) were isolated from the deep-sea-derived *Penicillium citreonigrum* MCCC 3A00169. The structures of the new compounds were determined by spectroscopic methods, including the HRESIMS, NMR, ECD calculations, and dimolybdenum tetraacetate-induced CD (ICD) experiments. Citreoviridins J−O (**1**–**6**) are diastereomers of 6,7-epoxycitreoviridin with different chiral centers at C-2–C-7. Pyrenocine A (**7**), terrein (**14**), and citreoviridin (**20**) significantly induced apoptosis for HeLa cells with IC_50_ values of 5.4 *μ*M, 11.3 *μ*M, and 0.7 *μ*M, respectively. To be specific, pyrenocine A could induce S phase arrest, while terrein and citreoviridin could obviously induce G0-G1 phase arrest. Citreoviridin could inhibit mTOR activity in HeLa cells.

## 1. Introduction

*Penicillium citreonigrum* is a commonly found fungus known for its contamination of rice with citreoviridin (CTV), a yellow mycotoxin related to the disease of acute cardiac beriberi [1]. In addition to CTV, *P. citreonigrum* can also produce other structurally diverse compounds, including azaphilones [2], chromones [3], alkaloids [3,4,5], sesquiterpenes [6,7], meroterpenes [2], etc. Most of these compounds showed cytotoxicity against different tumor cells. For example, 2-hydroxyl-3-pyrenocine-thio propanoic acid, a sulfur-containing polyketone isolated from a deep-sea-derived *P. citreonigrum* XT20-134, showed a potent effect against Bel-7402 tumor cells with an IC_50_ value of 7.6 μM [5]; sclerotioramine, a chlorinated alkaloid obtained from the terrestrial *P. citreonigrum*, exhibited moderate activity against the HepG2 cell line with an IC_50_ value of 7.3 μg/mL [3].

Due to their ability to produce new secondary metabolites, deep-sea-derived microorganisms have attracted more and more attention [8]. As part of our continuous investigations on deep-sea-derived fungi [9,10,11], *P. citreonigrum* MCCC 3A00169 was subjected for a systematic chemical study. Consequently, 6 new CTVs (citreoviridins J−O, **1**–**6**, Figure 1) and 22 known compounds (**7**–**28**, Appendix A) were obtained. We report herein the fermentation, isolation, structure, and bioactivities of these secondary metabolites.

## 2. Results and Discussion

The EtOAc extract of the fermentation broth of *P. citreonigrum* MCCC 3A00169 was subjected to extensive column chromatography (CC) over silica gel, ODS, and Sephadex LH-20. Final purification by semi-preparative HPLC yielded compounds **1**–**28**. By a comparison of the NMR and MS data with references, 22 previously reported components were identified as pyrenocine A (**7**) [12], cerebrosides A (**8**) [13], ergosterol (**9**) [14], *β*-sitosterol (**10**) [15], dehydrololiolide (**11**) [16], 3-hydroxy-4-methoxycinnamic acid (**12**) [17], terrein-d-glucoside (**13**) [18], terrein (**14**) [18], microsphaerone B (**15**) [19], surfactin C15 (**16**) [20], butyrolactone V (**17**) [21], microsphaerone A (**18**) [19], butyrolactone I (**19**) [22], citreoviridin (**20**) [23], isocitreoviridin (**21**) [24], (+)-cyclopenol (**22**) [25], tryptamine (**23**) [26], 1*H*-indole-3-carboxylic acid (**24**) [27], pyrenocine B (**25**) [28], citreo-*γ*-pyrone (**26**) [29], de-*O*-methyldiaporthin (**27**) [30], and 3,4-dihydro-3,4,8-trihydroxy-1(2*H*)-naphthalenon (**28**) [31].

Compound **1** was assigned a molecular formula C_23_H_30_O_7_ according to the protonated ion peak at *m/z* 419.2049 (calcd for C_23_H_31_O_7_, 419.2064) in its HR-ESI-MS spectrum (Appendix A), requiring nine double-bond equivalents. The ^1^H and ^13^C NMR spectroscopic data (Table 1 and Table 2, Appendix A) showed one methyl doublet and four methyl singlets, one methoxyl, ten methines (three oxygenated and seven olefinic carbons), and six quaternary carbons (three olefinic carbons and one carbonyl carbon). A close comparison of the NMR data for **1** with those of the citreoviridin I, a CTV isolated from a mangrove endophytic fungus *Penicillium* sp. BJR-P2 [32], indicated that they were structurally similar. Further analysis of their HR-ESI-MS data revealed that **1** was the dehydration product of citreoviridin I. The speculation was clearly evidenced by the HMBC correlations (Figure 2) from H_3_-21 (*δ*_H_ 1.35, s) to C-6 (*δ*_C_ 84.3)/C-7 (*δ*_C_ 77.2)/C-8 (*δ*_C_ 144.7), H-8 (*δ*_H_ 6.41, d, *J* = 15.2 Hz) to C-7 (*δ*_C_ 77.2), H-6 (*δ*_H_ 3.60, s) to C-8 (*δ*_C_ 144.7), and H_3_-20 (*δ*_H_ 1.23, s) to C-6 (*δ*_C_ 84.3).

The relative configuration of **1** was determined by coupling constants analysis and an NOESY experiment. *E*-configurations for three double bonds of ∆^8,10,12^ were identified on the basis of the large coupling constants of ^3^*J*_8,9_ (15.2 Hz), ^3^*J*_10,11_ (14.8 Hz), and ^3^*J*_12,13_ (15.2 Hz), respectively (Table 1). A cis relationship between H_3_-21 and H-6 was established for their distinct NOESY cross-peak (Figure 3). The relative configuration of the 2,3,5-trimethyl-tetrahydro-furan-3,4-diol in **1** was elucidated as 2*S**,3*S**,4*R**,5*S** according to the NOESY correlations of H-2 (*δ*_H_ 3.92, q, *J* = 6.4 Hz) with H_3_-19 (*δ*_H_ 1.29, s) and of H-4 with H_3_-19 and H-6. 

To determine the absolute configurations at C-6 and C-7 in **1**, ECD calculations of the optimized conformations of (6*R*,7*R*)-**1** (**1a**) and (6*S*,7*S*)-**1** (**1b**) were obtained at the B3LYP/6-31+G(d) level. The agreement between the calculated ECD spectrum of **1b** and the experimental ECD spectrum (Figure 4A) suggested a 6*S*,7*S*-configuration for compound **1**.

The absolute configuration of 2,3,5-trimethyl-tetrahydro-furan-3,4-diol residues was determined by a dimolybdenum tetraacetate [Mo_2_(OAc)_4_]-induced CD (ICD) experiment. Generally, 1, 2-diol compounds can react with Mo_2_(OAc)_4_ to form chiral complexes showing multiple Cotton effects in the range of 250–650 nm, of which the band near 310 nm is closely related to the absolute configuration of 1, 2-diol [33,34]. This method is also suitable for rigid cyclic 1, 2-diols [35,36]. In our experiments, the ICD spectrum of **1** exhibited a positive Cotton effect at 310 nm (Figure 5), suggesting a positive torsional angle for the O–C–C–O moiety. It was ascertained that the 3*S*,4*R*-form could maintain the favored conformation (Figure 1). Therefore, the absolute configuration of **1** was determined as (2*S*,3*S*,4*R*,5*S*,6*S*,7*S*)-6,7-dihydro-6,7-epoxycitreoviridin and named citreoviridin J.

An analysis of the ^1^H and ^13^C NMR spectroscopic data (Table 1 and Table 2) and HR-ESI-MS data of compounds **2**–**6** revealed that they share the same planar structure of **1**. However, minor differences in the ^13^C NMR chemical shift ranging from C-1 to C-7 indicated that **2**–**6** were diastereomers of 1 with different configurations in the 2,3,5-trimethyl-6-oxiranyl-tetrahydro-furan-3,4-diol residues. 

Therefore, similar to **1**, NOESY, ECD, and ICD experiments and coupling constants analysis were also employed to determine the absolute configurations of **2**–**6**. The geometry at *∆*^8,10,12^ double bonds in **2**–**6** was assigned as *E* based on the large coupling constants (*ca* 15 Hz). Likewise, the trans-configuration of epoxides and the cis-configuration for H-6 and H_3_-21 in **2**–**6** were assigned on the basis of the NOESY correlations of H-6 to H_3_-21 (Figure 3). The relative configurations of the 2,3,5-trimethyl-tetrahydro-furan-3,4-diol residues in **2**–**6** were also elucidated based on NOESY correlations, as shown in Figure 3. The ECD spectra of **2** and **5** showed positive Cotton effects at 215 nm and 373 nm and negative Cotton effects at 270 nm (Figure 4), which were in accordance with that of **1**. Therefore, 6*S*,7*S*-configuration was assigned for **2** and **5**. On the contrary, 6*R*,7*R*-configuration was assigned for **3**, **4**, and **6** because of the mirror-like ECD spectra (Figure 4). Similar to that of 1, the Mo_2_(OAc)_4_-induced CD spectra of **3** and **5** exhibited positive Cotton effects at 300–350 nm, indicating positive torsional angles for the O–C–C–O moieties (Figure 5). So, 3*S*,4*R*-configuration was defined for **3** and **5**. The negative Cotton effects at 300–350 nm represented negative torsional angles for the O–C–C–O moieties, establishing 3*R* and 4*S* configurations of **2**, **4**, and **6** (Figure 5). On the basis of the above evidence, the absolute configurations of **2**–**6** were defined as (2*R*,3*R*,4*S*,5*S*,6*S*,7*S*), (2*S*,3*S*,4*R*,5*S*,6*R*,7*R*), (2*S*,3*R*,4*S*,5*R*,6*R*,7*R*), (2*R*,3*S*,4*R*,5*R*,6*S*,7*S*), and (2*R*,3*R*,4*S*,5*S*,6*R*,7*R*), respectively, and were named as citreoviridins K–O.

All isolates were tested for antiproliferative activity against HeLa tumor cells. Compounds **7**, **14**, and **20** exhibited significant effects, with IC_50_ values of 5.4 μM, 11.3 μM, and 0.7 μM, respectively (Figure 6A). To further detect the apoptosis activity of these three compounds, Hela cells were analyzed by western blotting after treatment with **7**, **14**, and **20** for 40 h. The cleavage of PARP protein, a sensitive apoptotic marker, was used to detect the apoptosis activity. As shown in Figure 6B, they all induced potent apoptosis. It was reported that **20** could inhibit human umbilical vein endothelial cells (HUVECs) proliferation [37]. Compound **7** showed cytotoxicity against several cancer cells, with IC_50_ values ranging from 2.6 to 12.9 μM. Terrein (**14**) displayed strong cytotoxicities against human breast cancer MCF-7 cells [38] and human lung cancer A_549_ cells [39]. Hence, our findings were consistent with those reported in previous experiments, though different cancer cell lines were evaluated.

To detect their effect on cell cycle progression, HeLa cells were treated with compounds **7**, **14**, and **20** for 16 h, stained with propidium iodide, and analyzed by flow cytometry. As shown in Figure 7, **7** could induce S phase arrest, while **14** and **20** could obviously induce G0-G1 phase arrest. Compound **20** also inhibited the proliferation of HUVECs that were arrested at the G0/G1 phase [37].

As is known to all, the mTOR is one of the most usually activated signaling pathways in cancer. The major downstream target of the mTOR is the ribosomal protein S6.

Previously, citreoviridin induces myocardial apoptosis through the PPAR-γ-mTORC2-mediated autophagic pathway [40]. Therefore, compound **20** detected the protein level of the phosphorylation of S6. As is shown in Figure 8, compound **20** was found to obviously inhibit p-S6, indicating that **20** could strongly inhibit the mTOR pathway. Therefore, compound **20** might induce apoptosis through mTOR inhibition. 

## 3. Materials and Methods

### 3.1. General Experimental Procedures

Optical rotations were recorded on an Anton Paar MCP 100 polarimeter. ECD spectra were recorded on a Chirascan spectropolarimeter. The HRESIMS spectra were recorded on Q-Exactive Focus tandem mass spectrometry. The NMR spectra were recorded on a Bruker AVANCE III 400 MHz spectrometer. The preparative and semipreparative HPLC were performed on an Agilent Technologies 1260 infinity instrument using ODS or Chiralpak IC columns. UV spectra were recorded on a UV-8000 UV/Vis spectrometer. Column chromatography (CC) was performed on silica gel and Sephadex LH-20. The TLC plates were visualized under UV light or by spraying with 10% H_2_SO_4_.

### 3.2. Biological Material

The fungal strain *Penicillium citreonigrum* was isolated from the deep-sea sediment of the Northeastern Pacific at a depth of −2530 m. The voucher strain was preserved at the Marine Culture Collection of China (MCCC, Xiamen, China) and was given the accession number MCCC 3A00169.

### 3.3. Fermentation and Extraction

*P. citreonigrum* MCCC 3A00169 was grown under static conditions at 25 °C in 85 × 1 L Erlenmeyer flasks, each containing 80 g of rice and 120 mL of distilled H_2_O. After 47 days, the fermentation broth was extracted by EtOAc three times to give a crude extract (32 g).

### 3.4. Isolation and Purification

The EtOAc-soluble extract was subjected to CC over silica gel using gradient CH_2_Cl_2_-MeOH to give six fractions (Fr.1−Fr.6). Fraction Fr.1 (7 g) was separated into three subfractions (Fr.1-1−Fr.1-3) by CC over ODS with MeOH-H_2_O (40%→100%). Fr.1-1 (31 mg) was further separated by HPLC (10%→40%→100%, MeOH-H_2_O) to yield **7** (22 mg, *t*_R_ 32 min) and **11** (2 mg, *t*_R_ 29 min). Fr.1-3 (0.7 g) was purified by CC over Sephadex LH-20 (MeOH) to yield **9** (5 mg) and **10** (54 mg). Fraction Fr.2 (0.9 g) was separated by Sephadex LH-20 (MeOH) to yield **12** (2 mg). Fraction Fr.3 (10.3 g) was purified by CC over ODS with MeOH-H_2_O (30%→100%) to yield eleven subfractions (Fr.3-1−Fr.3-11). Fr.3-1−Fr.3-6 was directly separated by Sephadex LH-20 (MeOH) to yield **14** (295 mg), **22** (10 mg), **23** (2 mg), **24** (2 mg), and **16** (694 mg), respectively. Fr.3-7 was further purified to CC over Sephadex LH-20 (MeOH) and HPLC (55%→80%, MeOH-H_2_O) to yield **17** (5 mg) and a mixture of **5** and **6**, which was further isolated by the IC chiral column using n-hexane isopropyl alcohol (55%) as the mobile phase to yield **5** (2 mg, *t*_R_ 25.6 min) and **6** (2 mg, *t*_R_ 26.5 min). Fr.3-8 (877 mg) was purified by CC over Sephadex LH-20 (MeOH), HPLC (70%→85%, MeOH-H_2_O), and crystallization (MeOH) to afford compounds **15** (5 mg), **18** (3 mg), **20** (22 mg, *t*_R_ 27.5 min), and **21** (**5** mg, *t*_R_ 28.7 min). Fr.3-9 was separated by Sephadex LH-20 (MeOH) and HPLC (10%→40%, MeOH-H_2_O) to yield **26** (3 mg, t*_R_* 33 min). Fr.3-10 was separated by Sephadex LH-20 (MeOH) and HPLC (52%→70%, MeOH-H_2_O) to yield **3** (2 mg, t*_R_* 34 min), **4** (4 mg, t*_R_* 31.5 min), **19** (11 mg), and **27** (3 mg). Fr.3-11 was separated by Sephadex LH-20 (MeOH) and HPLC (55%→70%, MeOH-H_2_O) to yield **1** (8 mg, t*_R_* 33.2 min), **2** (7 mg, t*_R_* 31.5 min), and **25** (4 mg). Fraction Fr.4 (0.9 g) was subjected to CC over ODS with MeOH-H_2_O (20%→100%) and CC on Sephadex LH-20 (MeOH) to yield **28** (3 mg). Fraction Fr.5 (1.2 g) was separated by CC over Sephadex LH-20 (MeOH) and HPLC (5%→20%, MeOH-H_2_O) to yield **8** (70 mg) and **13** (2 mg, *t*_R_ 25 min).

*Citreoviridin J* (***1***): Yellow powder; [*α*${]}_{D}^{20}$ − 10 (*c* 0.10, MeOH); ECD (MeOH) *λ*_max_ (Δ*ε*) 372 (1.87), 271 (9.81), 216 (6.99) nm; UV (MeOH) *λ*_max_ (log *ε*) 196 (3.52), 273 (4.30), 369 (4.31) nm; ^1^H and ^13^C NMR data, Table 1 and Table 2; HRESIMS *m/z* 419.2049 [M + H]^+^ (calcd for C_23_H_31_O_7_, 419.2064).

*Citreoviridin K* (***2***): Yellow powder; [*α*${]}_{D}^{20}$ + 2 (*c* 0.10, MeOH); ECD (MeOH) *λ*_max_ (Δ*ε*) 373 (1.13), 271 (2.88), 217 (2.54) nm; UV (MeOH) *λ*_max_ (log *ε*) 196 (4.50), 274 (4.32), 369 (4.30) nm; ^1^H and ^13^C NMR data, Table 1 and Table 2; HRESIMS *m/z* 419.2063 [M + H]^+^ (calcd for C_23_H_31_O_7_, 419.2064).

*Citreoviridin L* (***3***): Yellow powder; [*α*${]}_{D}^{20}$ − 113 (*c* 0.10, MeOH); ECD (MeOH) *λ*_max_ (Δ*ε*) 362(3.40), 270 (9.35), 205 (6.88) nm; UV (MeOH) λ_max_ (log *ε*) 197 (3.68), 273 (4.40), 367 (4.41) nm; ^1^H and ^13^C NMR data, Table 1 and Table 2; HRESIMS *m/z* 419.2056 [M + H]^+^ (calcd for C_23_H_31_O_7_, 419.2064).

*Citreoviridin M* (***4***): Yellow powder; [*α*${]}_{D}^{20}$ + 20 (*c* 0.10, MeOH); ECD (MeOH) *λ*_max_ (Δ*ε*) 370 (0.70), 272 (8.06), 216 (5.83) nm; UV (MeOH) *λ*_max_ (log *ε*) 197 (3.67), 273 (4.45), 369 (4.43) nm; ^1^H and ^13^C NMR data, Table 1 and Table 2; HRESIMS *m/z* 419.2055 [M + H]^+^ (calcd for C_23_H_31_O_7_, 419.2064).

*Citreoviridin N* (***5***): Yellow powder; [*α*${]}_{D}^{20}$ − 52 (*c* 0.10, MeOH); ECD (MeOH) *λ*_max_ (Δ*ε*) 373 (0.58), 272 (−2.55), 231 (1.27) nm; UV (MeOH) *λ*_max_ (log *ε*) 198 (3.67), 272 (4.41), 369 (4.33) nm; ^1^H and ^13^C NMR data, Table 1 and Table 2; HRESIMS *m/z* 419.2045 [M + H]^+^ (calcd for C_23_H_31_O_7_, 419.2064).

*Citreoviridin O* (***6***): Yellow powder; [*α*${]}_{D}^{20}$ − 12 (*c* 0.10, MeOH); ECD (MeOH) *λ*_max_ (Δ*ε*) 367 (−1.11), 274 (5.18), 211 (−4.03) nm; UV (MeOH) *λ*_max_ (log *ε*) 197 (3.43), 274 (4.28), 368 (4.39) nm; ^1^H and ^13^C NMR data, Table 1 and Table 2; HRESIMS *m/z* 419.2042 [M + H]^+^ (calcd for C_23_H_31_O_7_, 419.2064).

### 3.5. ECD Calculation

The conformational analysis was first performed via random searching in the Stochastic using the MMFF94 force field with an energy cutoff of 7.0 kcal/mol and an RMSD threshold of 0.2 Å. All conformers were consecutively optimized at the PM6 and HF/6-31G(d) levels. Dominative conformers were re-optimized at the B3LYP/6-31G(d) level in the gas phase. The theoretical ECD spectra were calculated with the B3LYP method at the 6-311G(d,p) level in MeOH using Gaussian 09. The ECD spectrum was simulated by overlapping Gaussian functions for each transition [41].

### 3.6. Measurement of ICD Spectra

Compounds **1**–**6** were first dissolved in appropriate DMSO. Then, a quantity of Mo_2_(OAc)_4_ were added, with a ligand-to-metal ratio of approximately 1:1.2. The first CD spectrum (CD_0_) was recorded immediately after mixing and scanned every 10 min until a stationary CD spectrum (CD_1_) was measured. The induced CD (ICD) spectra were calculated from the CD of the ligand–metal complex (CD_1_) deducting the inherent CD (CD_0_).

### 3.7. The Antiproliferative Bioassay

As reported previously, the experiment was conduct using the Cell Counting Kit-8 (CCK-8) assay [42]. Briefly, HeLa cells were seeded in a 96-well plate at a density of 2000 cells/well and were cultured in MEM/EBSS (MEM) containing 10% FBS at 37 °C. After 24 h, the cells were treated with the test compounds, and incubation continued for 72 h. Then, 10 μL CCK-8 solution was added to each well. After incubation at 37 °C for 4 h, the absorbance value of each well was determined using a multi-well plate reader at 450 nm. 

### 3.8. Flow Cytometry

After the indicated time treatment, cancer cell arrest was assessed by an FACScan flow cytometer (Beckman Coulter, California, USA), following the manual procedure. The cells were harvested by trypsin digestion, washed with PBS, and fixed with ice-cold 70% ethanol at 4 °C overnight. The fixed cells were then washed twice in PBS and treated for 30 min at RT with propidium iodide in PBS and analyzed. Flow cytometry data were analyzed using CytExpert (Beckman Coulter).

### 3.9. Western Blot Analysis

For Western blot assays, HeLa cells were treated with the compounds for the indicated time. Then, the cells were harvested, lysed, and centrifuged at 12,000 rpm/min for 10 min. The supernatant was added with a 1/5 volume of 5 × SDS and boiled. After electrophoresis, protein samples were transferred to the PVDF film, blocked with fat-free milk, incubated with the first antibody and washed, and then incubated with the secondary antibody and washed. Then, the ECL droplets were reacted on the membrane surface, and the bands were imaged by the multifunctional chemiluminescence imaging system.

## 4. Conclusions

From the deep-sea-derived *Penicillium citreonigrum* MCCC 3A00169, 6 new and 22 known compounds were obtained. Compounds **7**, **14**, and **20** significantly induced apoptosis against HeLa cells with IC_50_ values of 5.4 μM, 11.3 μM, and 0.7 μM.

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
