# Peer review of "Chemical Constituents of the Deep-Sea-Derived Penicillium citreonigrum MCCC 3A00169 and Their Antiproliferative Effects"

_marinedrugs, 2022, doi:10.3390/md20120736_

Round 1

Reviewer 1 Report

The dimolybdenum tetraacetate induced CD (ICD) experiment (used by the authors) consists of complexing molybdenum tetraacetate by a ligand and then using the molybdenum moiety as a chromophore in CD spectroscopy. Rhodium (II) tetraacetate and tetrakistrufluoroacetate have been used similarly (Frelek et al.). Unlike NMR spectroscopy, these methods are less versatile and are used in a relatively narrow range. Therefore, explaining the technique briefly in the text would be helpful and also include some references, for example Frelek et al, Chirality 1998, 578; Liu et al., Chin. J. Org. Chem., 2010, 30, 1270; Frelek et al., J. Org. Chem.  2013, 72, 2906; G. Pescitelli, 2004...

Reviewer 2 Report

The manuscript describes the structure elucidation of six new members of the mycotoxin citreoviridin derivatives together with 21 known compounds isolated from the marine Penicillium citreonigrum...

The work deals mainly with structural analysis, Absolute Configuration assignment of the new compounds and the biological activities of all isolated compounds.  While the new compounds were biologically not active, 3 known compounds 7, 14 and the very known toxin citreoviridin (20) were found active at miromolar level against HeLa tumor cell.

This reviewer suggests accepting the manuscript after addressing the following points:

Comments:

The planar structure arguments are good, but the concern of this reviewer is the more or less rigor in the reasoning leading to the assigned relative configurations and the confrontation of the biological results described in this work with the previously published studies regarding cytotoxicity and apoptosis of the same compounds.

This two aspects should be improved with more evidences.

1) Since the title does not indicate whether the antiproliferative activity is due to one of the 6 new metabolites (none are active), it does not reflect the content of the article. And moreover, the structures of the known active products are not given in the manuscript?? please adjust the content to the title and the abstract.

2) Please cite in good time the previous publications describing the same (or very similar) compounds as those described in the article. Please compare them with the methods and results of the article. Although the article seems concise and easy to read, it is not acceptable when the state of the art is ignored, even if it is not intended. 2 Examples:

- For structural determination using the same approach for very similar compounds (citreoviridin H and citreoviridin I), reference https://doi.org/10.3390/md20090583, is not cited!

- For antiproliferative biological activity, G0/G1 phase arrest of the Citreoviridin and derivatives: reference http://dx.doi.org/10.1016/j.etap.2014.02.016; is not cited!!

- For apotosis and mTOR pathway involvement, reference https://doi.org/10.1016 /j.cbi.2019.108795  is not cited  Etc….

3) Why the new series of names penicitreoviridins A−F, and not citreoviridin J, K, L, M etc… following the very recent article (https://doi.org/10.3390/md20090583)

4) Structures of the biologically active compounds Pyrenocine A (7), terrein (14), and citreoviridin (20) should be given even if they are known. In addition, citreoviridin (20) is used to support the structural determination by comparison. its stricture and NMR should be included in Tables 1 and 2 for comparison.

5) Page 3, last paragraph: the epoxide is not trans but cis (in any case, theose drawn in Figures 1 and 3). The reference 32 given to argue the trans configuration is useless here because the products do not have the same configuration. Please delete it.

6) Page 6, first paragraph: All the epoxides are not trans but cis.

Please check and correct.

Related to this point, the relative configurations of the new compounds 1-6 should be cheked and additional NOESY correlations must be added in Figure 6 to ensure conformational and configurational elements. The given correlations are not sufficient to correctly conclude on the conformation and configurations of the relative compounds.

This reviewer is really asking the question if compounds 4 and 5 are not like 1 and 3, then 2 and 6, which have the same configurations on the tetrahydrofuran part and only the epoxide is above or below the plane?? This gives three pairs of compounds generated by the late epoxidation. The point merits to be checked carefully.

7) page 7, first paragraph: micromolar activity could not qualified as potent!!

8) page 7, Last paragraph:

compound 20 was found to obviously inhibit p- S6, indicating 20 could strongly inhibit mTOR pathway (Figure 8). Therefore, compound 20 might induce apoptosis through mTOR inhi-bition… :

please refer to the publication (https://doi.org/10.1016 /j.cbi.2019.108795) not cited in the manuscript. It describes similar results for the same compound!!

9) 13C NMR of compounds 5 and 6 (2 mg each) are missing in the SI

Reviewer 3 Report

The manuscript entitled “Chemical Constituents of the Deep-Sea-Derived Penicillium citreonigrum MCCC 3A00169 and Their Antiproliferative Effects” described the discovery of six new compounds (penicitreoviridins A−F, 16) and 22 known compounds. Some of the known isolates (7, 14, and 20) induced the apoptosis for Hela cells at low concentrations. The paper was well-written, and the structural elucidation was solid. Thus, I suggest that this manuscript may be suitable for publication in Marine Drugs after minor revision.

Here are some comments about the manuscript, which should be addressed by the authors.

1.    Please provide the chemical structures of the known compounds in SI. Since some of the known compounds showed potent activity, authors should provide evidence for elucidating the structures in SI (1H NMR, 13C NMR, MS, and optical rotation).

2.    For Figure 5, better line up the ICD spectra with pure compounds as controls. Details see J. Org. Chem. 2019, 84, 5, 2568–2576, Figure 6.

3.    Please provide retention time during HPLC purification in section 3.4
